# App-Based Mindfulness Meditation Training and an Audiobook Intervention Reduce Symptom Severity but Do Not Modify Backward Inhibition in Adolescent Obsessive-Compulsive Disorder: Evidence from an EEG Study

**DOI:** 10.3390/jcm12072486

**Published:** 2023-03-24

**Authors:** Sarah Rempel, Lea L. Backhausen, Maria McDonald, Veit Roessner, Nora C. Vetter, Christian Beste, Nicole Wolff

**Affiliations:** 1Department of Child and Adolescent Psychiatry, Faculty of Medicine of the TU Dresden, 01307 Dresden, Germany; sarah.rempel@ukdd.de (S.R.);; 2Cognitive Neurophysiology, Department of Child and Adolescent Psychiatry, Faculty of Medicine of the TU Dresden, 01307 Dresden, Germany; 3Department of Psychology, MSB Medical School Berlin, 14197 Berlin, Germany

**Keywords:** adolescents, app, backward inhibition, EEG, intervention, OCD, mindfulness meditation

## Abstract

(1) Background: 1–2% of children and adolescents are affected by Obsessive-Compulsive Disorder (OCD). The rigid, repetitive features of OCD and an assumed disability to inhibit recent mental representations are assumed to have led to a paradoxical advantage in that the Backward Inhibition (BI) effect was recently found to be lower in adolescents with OCD as compared to healthy controls. It was hypothesized that app-based mindfulness meditation training could reduce the disability to inhibit recent mental representations and thus increase the BI-effect by adapting cognitive flexibility and inhibition abilities according to healthy controls. (2) Methods: 58 adolescents (10–19 years) with OCD were included in the final sample of this interviewer-blind, randomized controlled study. Participants were allocated to an intervention group (app-based mindfulness meditation training) or an (active) control group (app-based audiobook) for eight weeks. Symptom (CY-BOCS), behavioral (reaction times and mean accuracy), and neurophysiological changes (in EEG) of the BI-effect were analyzed in a pre-post design. (3) Results: The intervention and the control group showed an intervention effect (Reliable Change Index: 67%) with a significant symptom reduction. Contrary to the hypothesis, the BI-effect did not differ between pre vs. post app-based mindfulness meditation training. In addition, as expected the audiobook application showed no effects. Thus, we observed no intervention-specific differences with respect to behavioral (reaction times and mean accuracy) or with respect to neurophysiological (perceptual [P1], attentional [N1], conflict monitoring [N2] or updating and response selection [P3]) processes. However, in an exploratory approach, we revealed that the BI-effect decreased in participants who did not benefit from using an app, regardless of group. (4) Conclusions: Both listening to an app-based mindfulness meditation training and to an audiobook reduce symptom severity in adolescent OCD as measured by the CY-BOCS; however, they have no specific effect on BI. The extent of the baseline BI-effect might be considered as an intra-individual component to predict the benefit of both mindfulness meditation training and listening to an audiobook.

## 1. Introduction

Obsessive-Compulsive Disorder (OCD) is one of the most frequent mental disorders. It affects 1–2% of children and adolescents [1] and is associated with a poor quality of life [2]. OCD is characterized by persistent intrusive thoughts (obsessions) and/or repetitive behaviors (compulsions) often aimed at neutralizing anxiety or preventing a dreaded event [1]. Particularly difficult for patients with OCD is the suppression and detachment of unpleasant thoughts and actions. One assumption is that impaired executive functioning (EF), in particular deficits with respect to inhibition [3,4,5] and cognitive flexibility [3,6], may be underlying mechanisms of OCD-related symptoms [7,8]; i.e., maladaptive patterns of inhibitable, repetitive cognitions and behaviors. In psychological tasks, inhibition is critical for suppressing ineffective or irrelevant mental representations of tasks [9]. This ability is fundamental for cognitive flexibility that, in turn, is required, e.g., to switch quickly between different tasks; i.e., also from an actual, perhaps dominant to another new mental representation [9].

Inhibition and cognitive flexibility can be examined in task switching paradigms [10,11]. In these paradigms, the inhibition of the mental representation of the previous task A is needed to switch to task B to increase cognitive flexibility. In, for example, an A_1_→B→A_2_ task sequence, however, after the successful inhibition of the mental representation of task A_1_ in order to switch to task B the further inhibition of the mental representation of task A_1_ must be abandoned to optimize the switch to task A_2_. The latter process is defined as backward inhibition (BI). In this context, the BI-effect, also known as n-2 repetition costs [12], is a further commonly used parameter. It can only be measured in a task switching paradigm involving switching minimally between three tasks (A, B, C) [13,14]. To assess the BI-effect two conditions are required: (1) the n-2 task repetition condition (e.g., A_1_→B→A_2_ task sequence), also called BI –condition, and (2) the n-2 task switch condition (e.g., C→B→A or B→C→A task sequence) also called the baseline (BASE)-condition. The BI-effect is defined as the contrast of the BI condition and the BASE condition [13,15] and reflects the effect of an n-1 task switch on the nth trial, with the latter reflecting the same condition (rule) as the n-2 trial. A smaller BI-effect indicates a more effective BI.

Wolff et al. [11] recently reported a smaller BI-effect in adolescents with OCD than in healthy controls paralleled by a larger P1 event related potential (ERP) amplitude in the electroencephalography (EEG). As the P1 was considered to reflect the process of inhibiting relevant and irrelevant information [16], the observed effects in the work of Wolff et al. [11] may suggest an increased processing of and engagement with previous stimuli and thus no inhibition of recent now irrelevant trials, which were otherwise abandoned (inhibited) in healthy controls. Wolff et al. postulated that the rigid, repetitive features of OCD and an assumed disability to inhibit recent mental representations lead to this paradoxical advantage in BI paralleled by a larger P1 ERP amplitude [11]. The faster reaction times (RTs) in the BI condition with a n-2 task repetition (e.g., A_1_→B→A_2_ task sequence) compared to healthy controls might in some way reflect the repetitive behavior of OCD. The reduced ability to inhibit mental representations in a proper way leads to a constant mental presence of the penultimate task resulting in better BI, but at the cost of higher cognitive load. This might also be explained by being stuck in the past rather than paying attention to the present moment, i.e., being mindful.

Therapeutic interventions, such as mindfulness-based therapies, may thus be the method of choice when treating patients with OCD. In recent years, third wave CBT, including mindfulness-based techniques have become more widely utilized as a supplemental or stand-alone therapy option for various mental health conditions [17]. Mindfulness can be defined as consciously paying attention, focusing on the present moment, and thus actively inhibiting the past, the future, or ruminating thoughts. Mindfulness has been stated to involve inhibition of thoughts and mental representations that were previously relevant but are not any longer in the present moment [18]. Therefore, assuming that an underlying mechanism of mindfulness is to induce a specific meta-control state [19,20], with a sharpened focus on one specific goal including the processing of relevant information (e.g., processing of new rules, the inhibition of recently abandoned mental representations), it can be hypothesized that a mindfulness meditation training in OCD, may (paradoxically) increase the BI-effect through both, increased inhibition abilities as well as cognitive flexibility seen in healthy controls. In this respect, both the phenomenology of OCD (rigid, repetitive cognition, and behavior) [21] and differences in the BI-effect in children and adolescents with OCD compared to healthy controls [11] suggest a mindfulness-based approach to be a promising intervention. To our knowledge, only one single case study [22] reported an eight-week mindfulness-based treatment for adults with OCD resulting in clinically significant reductions of OCD symptoms. Aside from this, studies on other forms of personalized and adapted mindfulness-based interventions in adults with OCD have also been conducted and support the efficacy of mindfulness-based treatment programs at the symptom level [23,24,25,26,27,28,29,30,31,32,33,34,35]. In summary, to the best of our knowledge no research on the efficacy of mindfulness-based intervention on children and adolescents with OCD had been conducted, and numbers of studies on neurophysiological effects of mindfulness meditation training fall far behind the extent of pertinent clinical research.

Thus, the present interviewer-blind, randomized, and actively controlled study aimed to investigate the efficacy as well as underlying neurophysiological mechanisms of app-based mindfulness meditation training in a sample of children and adolescents with OCD. Thereby, we aimed to test the hypothesis of whether learning specific techniques for focusing, reassessing, and refocusing ruminating thoughts, as well as for tensing and relaxing (to name a few), as taught in mindfulness meditation training, better enables children and adolescents with OCD to successfully inhibit mental representations. In addition to the intervention group (app-based mindfulness meditation training), an active control group was surveyed. They also listened to auditory, verbally mediated content via an audiobook app. In contrast, in this group no specific mindfulness techniques as mentioned above were addressed; instead, they listened to some chapters of the audiobook *Harry Potter and the Philosopher’s Stone and Harry Potter and the Chamber of Secrets*. Although listening to an audiobook possibly establishes concentration and relaxation impulses, which thus may (briefly) contribute to a reduction of OCD symptoms, we did not expect that listening to an audiobook would influence inhibition processes sustainably.

The clinical outcome measure was symptom-related, measured via the score on the German Children’s Yale-Brown Obsessive-Compulsive Scale (CY-BOCS), which is considered the gold standard for the assessment of OCD severity [36]. Symptom (CY-BOCS), behavioral (reaction times and mean accuracy), and neurophysiological changes using an EEG were analyzed in a pre-post design. Based on previous findings [11], we expected that inhibition improvements would be reflected by increased RTs during the BI condition compared to the BASE condition resulting in an increased BI-effect in the intervention group. This would reflect an adaptation to the pattern of healthy controls (see Wolff et al. [11]). Based on previous findings [11,15,37], we expected effects primarily in the early perceptual (P1) and attentional (N1) ERP.

## 2. Materials and Methods

### 2.1. Participants

We included 76 adolescents (10–19 years) between May 2019 and May 2022 (see Table 1) in our interviewer-blind, randomized, and actively controlled study to examine the effectiveness and underlying mechanisms of app-based mindfulness meditation training for individuals with OCD. Outpatients with a confirmed OCD diagnosis (ICD-10 diagnoses F42.0, F42.1, F42.2, F42.8, or F42.9) and individuals with a suspected OCD and a CY-BOCS score ≥ 8 points were included (see Table 1). Exclusion criteria were: neurological, developmental, or other psychiatric disorders that are being treated primarily, a history of psychosis, a current severe depressive episode, a current substance use disorder, acute suicidal tendencies, or an intelligence quotient < 70 (IQ level 5–8: mental retardation according to the German multi-axial-classification system [38]) administered by “Der Zahlen-Verbindungs Test” (ZVT) [39] (English: number-connection test). In addition, during study participation, no change of any additional therapeutic intervention was allowed, i.e., possibly existing/enduring therapies (e.g., medical, physio-, ergo- or behavioral therapy) had to remain constant during the entire study period and new therapies could only be started after the end of participation in the study. In addition, 18 participants were excluded from the analyses because of (a) a high error rate (less than 15% correct in any condition of the BI paradigm, N = 5), (b) technical difficulties during the EEG measurement (N = 2), (c) low compliance with using the app (more than 2*SD away from M of the group regarding total minutes, N = 3), (d) time or appointment issues because of the COVID-19 pandemic (N = 2), and (e) dropouts (N = 6). The final sample consisted of 58 adolescents (M_age_ = 15.54 years ± 2.09, range: 10.73–19.59 years; M_IQ_ 103.1 ± 14.22, range: 79–139; 43.1% male; see Table 1). All participants had normal or corrected to normal vision. We used a semi-structured interview according to the DSM-V and current clinical reports to assess possible psychiatric comorbidities. Comorbid diagnoses were present in 37 of the final sample of 58 participants, and 20 participants took medication (see Figure 1).

### 2.2. Intervention and Procedure

After they consented to being contacted for potential research inquiries, participants were recruited from the Department of Child and Adolescent Psychiatry and Psychotherapy at the University Hospital Dresden, Germany. After inclusion, participants were randomly allocated to an intervention group (app-based mindfulness meditation training) or (active) control group (audiobook app). All participants were assigned a de-identified study ID to maintain a blind study. There were no significant differences in terms of demographic characteristics or pre-treatment (medication, treatment status) between both groups (all *p* > 0.05, refer to Table 1). All participants received two EEG recording appointments (a pre-appointment and, following the 8-week use of the apps, a post-appointment) to collect EEG data while performing the BI paradigm and another task switching paradigm [42]. The order of performance of the two paradigms was randomized. However, the other task switching paradigm results are not of interest in this paper. The intervention group received the app ‘7Mind’ with the following meditation courses: “Kids & school (Kids & Schule)”, “Basics (Grundlagen)”, “Basics intensive 1 (Grundlagen Intensiv 1)”, “Basics intensive 2 (Grundlagen Intensiv 2)”, “Basics intensive 3 (Grundlagen Intensiv 3)”, “Basics intensive 4 (Grundlagen Intensiv 4)”, “luck (Glück)”, “gratitude (Dankbarkeit)”, and “sleep (Schlaf)”. The app can be downloaded in the Google play store or App store (https://www.7mind.de/download, accessed on 19 December 2022). Each audio mindfulness meditation course contains seven practice sessions of seven–twelve minutes each. Participants were asked to practice one mindfulness meditation during the day and one when going to bed (sleep meditation) as outlined by the structured intervention plan. A calm male voice gives instructions, follows the same structure, and thus makes it easier for the participant to internalize the dynamics of the meditation process more and more deeply. The active control group used the app Audible (https://www.audible.de/ep/apps, accessed on 19 December 2022) with the audiobooks *Harry Potter and the Philosopher’s Stone* (German: *Harry Potter und der Stein der Weisen*) for the first 4 weeks and *Harry Potter and the Chamber of Secrets* (German: *Harry Potter und die Kammer des Schreckens*) for the remaining 4 weeks, as also outlined by the structured intervention plan. Both groups used the app on a smartphone for 8 weeks, twice a day. The intervention group used the app for 813 min in total (±279), compared with 1081 min (±284) in the active control group. Both groups followed a structured listening plan, given to them at their first appointment. While using the apps, participants remained in contact with the study team via a smartphone messenger service to maintain compliance. Participants also had an interim appointment with the study team after four weeks to discuss their progress. The CY-BOCS and the Zwangsinventar (ZWIK) (English: obsessive-compulsive inventory), which is a widely used self (ZWIK-S) and parent-rating (ZWIK-P) questionnaire [41], were administered. All standardized interviews and questionnaires for the participants and their parents were administered at the pre-and post-appointment.

### 2.3. Backward Inhibition Task

To measure backward inhibition in task switching, the present study used a BI paradigm that was put forth by Mayr and Keele [13]. We applied the modified version of Koch et al. [14], that has been used in other investigations [11,15,43,44,45]. Participants performed the BI paradigm before and after using the app (pre- and post-appointment). During the task, participants sat in front of a computer monitor with a viewing distance of approximately 60 cm in the EEG laboratory of the Department of Child and Adolescent Psychiatry of the University Hospital Dresden. The task was presented via the software “Presentation” (Neurobehavioral System, Inc., 20.3., Berkeley, CA, USA). Written instructions were presented on the screen and verbally explained to the participants. Participants were encouraged to respond as quickly and accurately as possible. The experimental paradigm is shown schematically in Figure 2. Participants’ response input was made by pressing the left or right control key on a QWERTZ keyboard. Stimuli were presented in white on a black computer screen. Each trial began with the presentation of a cue for 100 ms. Geometric figures presented the cues. The cues were a square (rule odd or even = rule A), a rhombus (rule smaller or larger than 5 = rule B), or a triangle (rule press both control keys simultaneously = rule D). Afterward, the target stimuli were presented as a number between 1 and 9 within the cue, excluding the number 5. The cue and target stimulus were shown until the participant responds (see Figure 2). During rule D, unlike the other two rules, participants were asked to press both control keys simultaneously as quickly as possible (within 1000 ms). After response input, there was an interval of 1500 ms until the next cue is presented. Within this interval, a fixation cross was shown in the center of the screen. If participants did not follow these speed cues, the German word “Schneller” (English: “Faster”) appeared above the stimuli to alert participants to speed up their response input. If the participants did not respond within the 1000 ms or after the speed-up cue has faded out, they received the feedback “Too slow!”. Too late responses or not pressing the two control keys simultaneously (50 ms) during rule D are errors. If a response is incorrect, the German word “Falsch! (English: “Incorrect!”) was displayed on the screen for 500 ms in all task conditions. The paradigm comprises 768 trials, divided into 8 blocks. At the end of each block, the mean RT of the previous block was displayed. At this point, participants also had the opportunity to take a short break. Each cue, target stimuli, and possible combination of the two occurred with the same frequency. It was also ensured that the target stimuli in the current trial was different from the target stimuli in the last trial with the same cue. Each trial forms a sequence, i.e., a three-way combination of the three possible rules. All possible sequence combinations (ABA; ABD; ADA; ADB; BAB; BAD; BDA; BDB; DAB DAD; DBA; and DBD) are equally frequent. Sequences with the same cue in the last trial as n-2 trials before are categorized as back-switching sequences (BI condition). Trials where this is not the case, are categorized as baseline sequences (BASE condition). To ensure that the participants understand the task, a practice block with a total of 12 trials was prepended at the pre- and post-appointment. For this study, we used the following sequences as they have been reported to show the strongest BI-effect [11,15,45]: ABA, BAB, DBA, and DAB.

### 2.4. EEG Recording and Analysis

The EEG data was recorded with 500 Hz using 60 Ag/AgCl electrodes dispersed evenly over an elastic cap (ground electrode at coordinates θ = 58, ϕ = 78; reference electrode at coordinates θ = 90, ϕ = 90; electrode impedances below 20 kΩ). The Brain Vision Analyzer 2.1 was used to analyze the EEG data that had been saved offline (Brain Products GmbH, Gilching, Germany). The captured data was initially downsampled to 256 Hz and band-pass filtered from 0.5 to 20 Hz (slope of 48 db/oct each) before being processed. A common average reference was used to re-reference all channels. A manual raw data inspection was conducted to remove technical artifacts and breaks. In contrast, periodically artifacts (such as pulse artifacts, horizontal and vertical eye movements) were subsequently detected and corrected by an independent component analysis (ICA; infomax algorithm). Previously removed channels were interpolated when necessary. After these corrections, data were segmented for every condition separately.

Only trials with correct responses were included. A baseline correction was then set to a time interval from −300 ms to 0 ms before the segments were averaged for each condition. The cue onset was set to time point 0 and ended 2000 ms after that, resulting in an overall segment length of 2300 ms. Afterward, an automated artifact rejection procedure was applied: an activity below 0.5 μV in a 100 ms period and a maximal value difference of 200 μV in a 200 ms interval were used as rejection criteria. Next, a current source density (CSD) transformation was run. The segmented data for the BASE condition were trial sequences of DBA and DAB. For the BI condition, we used trial sequences ABA and BAB. The P1, N1, N2 and P3 ERP component amplitudes were quantified on the single subject level. The time windows and electrode sites for data (amplitude) quantification were chosen based on previous studies [11]. The P1 and N1 were measured at P7 and P8 following the cue (c-P1: 90–100 ms; c-N1: 160–195 ms) and following the target, which was added after a SOA of 100 ms (t-P1: 250–290 ms; t-N1: 340–360 ms). At electrode Cz, the N2 was quantified in the time window from 400 to 430 ms. The P3 was quantified at electrodes PO1 and PO2 in the time interval from 570 to 610 ms.

### 2.5. Statistics

We used IBM SPSS Statistics (Version 28.0.1.1) to analyze the data. The symptomatology data was analyzed by mixed effects repeated measure ANOVAS with the within-subject factor “Appointment” (pre vs. post) and between-subject factor “Group”. With regards to the BI-paradigm, we removed the first two trials of each block, all trials with an error, and the two trials following an error from both the behavioral and neurophysiological data. We eliminated trials with RTs greater than 2500 ms or less than 100 ms from the remaining trials. The mean number of the remaining trials for each sequence that entered the analysis was above 39 for all sequence conditions (ABA PRE: 59.53 ± 5.91; BAB PRE: 59.12 ± 6.59; DAB PRE: 60.5 ± 6.62; DBA PRE: 59.74 ± 5.67; ABA POST: 61.12 ± 4.53; BAB POST: 62.36 ± 4.36; DAB POST: 61.33 ± 4.67; DBA POST: 62.22 ± 5.1).

Behavioral and neurophysiological data were analyzed using mixed effect ANOVAs. These models included the within subject factor “Condition” (backward inhibition/BI vs. baseline/BASE), “Appointment” (pre- vs. post-appointment) and the between-subject factor “Group”. For the neurophysiological data, the factor “Electrode” was included as additional within-subject factor whenever necessary. Greenhouse–Geisser corrections were applied. All post-hoc tests were Bonferroni-corrected.

Finally, Bayesian analyses were performed by Masson’s recommendation [46] to determine whether the null (H_0_) or alternative hypothesis (H_1_) was more plausible in the event of a non-significant main or interaction effect of “Group”. In short, this method returns the likelihood of the H_0_ and H_1_, given the obtained data.

## 3. Results

### 3.1. Symptom Data

#### 3.1.1. CY-BOCS

Intervention vs. Control group. CY-BOCS scores for the intervention and control group can be found in Table 1. The repeated measures ANOVA revealed a significant effect on the CY-BOCS score for “Appointment” (pre vs. post) [F(1,56) = 30.679; *p* < 0.001; η^2^ = 0.354], showing that the score was higher at the pre-appointment (17.41 ± 0.88) than at the post-appointment (13.32 ± 0.85). There was no interaction effect with “Group”; F(1,56) = 0.102; *p* = 0.751; η^2^ = 0.002. Comparisons of the mean between the pre- and post-appointment in the intervention- and control group can be found in Table 1.

The effect of the app-application was represented in terms of Cohen’s d, a standardized measure of effect size. Cohen’s d [47] is defined as
$$d=\frac{\left(M_{1}-M_{2}\right)}{{\mathrm{SD}}_{\mathrm{pooled}}}$$
where M_1_ and M_2_ represent means of pre- and post-appointment scores and SD the pooled standard deviation. Using the above formula, we obtained an effect size of d = 0.61, which represents a medium effect size. We examined the proportion of participants who improved to assess the clinical significance of treatment. Clinical significance was represented in terms of the Reliable Change Index (RC) [48] defined as
$$\mathrm{RC}=\left(X_{1}-X_{2}\right){\mathrm{SE}}_{\mathrm{diff}}$$
where X_1_ represents a participant’s pre-appointment score, X_2_ represents that same participant’s post-appointment score, and SE_diff_ is the standard error of difference between the two CY-BOCS scores. When RC exceeds 1.96, the post-appointment score likely reflects real change according to Jacobson and Truax [48]. In our sample, 39 (N = 20 of the intervention group, 19 of the control group) of the 58 participants (67.24%) had an RC above 1.96, suggesting that the change observed in these participants shows more than the fluctuations of an imprecise measuring instrument and reflects a clinically significant reduction of OCD symptoms. The proportion of participants with a RC above 1.96 did differ from the participants with a RC below 1.96, X^2^ (1, N = 58) = 6.9, *p* = 0.009.

#### 3.1.2. ZWIK-S

Intervention vs. Control group. ZWIK-S scores for the intervention and control group can be found in Table 1. The repeated measures ANOVA revealed a significant effect on the ZWIK-S score for “Appointment” (pre vs. post) [F(1,54) = 15.785; *p* < 0.001; η^2^ = 0.226], showing that the score was higher at the pre-appointment (41.88 ± 3.87) than at the post-appointment (31.88 ± 3.13). There was no interaction effect with “Group”; F(1,54) = 1.454; *p* = 0.223; η^2^ = 0.026. Comparisons of the mean between the pre- and post-appointment in the intervention- and control group can be found in Table 1. Regarding the RC (as explained in Section 3.1.1) 38 of 57 (66.67%) have a higher score than 1.96 (N = 20 in the intervention group, N = 18 in the control group) and reflect a clinically significant reduction of OCD symptoms. The proportion of participants with a RC above 1.96 did differ from the participants with a RC below 1.96 [X^2^ (1, N = 56) = 7.143, *p* = 0.008].

#### 3.1.3. ZWIK-P

Intervention vs. Control group. ZWIK-P scores for the intervention and control group can be found in Table 1. The repeated measures ANOVA revealed a significant effect on the ZWIK-E score for “Appointment” (pre vs. post) [F(1,55) = 8.994; *p* = 0.004; η^2^ = 0.141], showing that the score was higher at the pre-appointment (32.31 ± 2.45) than at the post-appointment (25.07 ± 2.29). There was no interaction effect with “Group”; F(1,54) = 0.141; *p* = 0.708; η^2^ = 0.003. Comparisons of the mean between the pre- and post-appointment in the intervention- and control group can be found in Table 1. Regarding the RC (as explained in Section 3.1.1) 35 of 57 (61.4%) have a higher score than 1.96 (N = 18 in the intervention group, N = 17 in the control group) and reflect a clinically significant reduction of OCD symptoms. The proportion of participants with a RC above 1.96 did not differ from the participants with a RC below 1.96 [X^2^ (1, N = 57) = 2.965, *p* = 0.085].

### 3.2. Backward Inhibition Paradigm

#### 3.2.1. Behavioral Data

Intervention vs. Control group. For the two-group comparison, the repeated measures ANOVA revealed a main effect of “Appointment” showing that RTs were higher during the pre-appointment (954.4 ms ± 31.52) than during the post-appointment (767.1 ms ± 24.42), F(1,56) = 106.286; *p* < 0.001; η^2^ = 0.655. There was another main effect of “Condition” showing that RTs were higher in the BI condition (886.54 ms ± 27.3) than in the BASE condition (834.97 ms ± 26.33), F(1,56) = 97.512; *p* < 0.001; η^2^ = 0.635 (Figure 3). No significant interactions were found (all F ≤ 1.074, *p* ≥ 0.305). Regarding the BI-effect (BI−BASE) the repeated measures ANOVA with the factors “Group” and “Appointment” revealed no significant effects (all F ≤ 1.074; all *p* ≥ 0.305).

Repeated measures ANOVA on performance accuracy revealed an interaction effect of “Condition” × “Appointment” [F(1,56) = 15.685; *p* < 0.001; η^2^ = 0.219] with a higher accuracy during the post-appointment in the BASE condition compared to the pre-appointment (64.26% ± 2.21 vs. 58.76% ± 2.22, *p* < 0.001). There was no corresponding effect in the BI condition (61.69% ± 2.59 vs. 61.56% ± 2.28, *p* = 0.94). There were no other significant effects (all F ≤ 3.683; all *p* ≥ 0.06). Regarding the BI-effect (BI condition−BASE condition) the repeated measures ANOVA with the factors “Group” and “Appointment” revealed a significant effect of “Appointment” [F(1,56) = 15.685; *p* < 0.001; η^2^ = 0.219]. Post hoc-tests showed that the BI-effect was significantly higher during the pre-appointment compared to the post-appointment (2.8% ± 1.3 vs. −2.57% ± 1.21). There was no effect of “Group” or an interaction with “Group”. (All F ≤ 2.504; all *p* ≥ 0.119). Given the non-significant interaction of “Condition” × “Appointment” in the RT as well as non-significant correlations between accuracy and RT in both conditions which were separately calculated for pre- and post-appointment (all *p* ≤ 0.19), a speed-accuracy tradeoff can be excluded.

#### 3.2.2. Neurophysiological Data

The analysis of the cue P1 (c-P1) component revealed a two-way interaction effect of “Electrode” × “Group” [F(1,56) = 4.477; *p* = 0.039; η^2^ = 0.074], where the amplitude of the electrode P7 was significant lower (14.85 µV ± 2.55) compared to the electrode P8 (21.57 µV ± 3.22, *p* = 0.029) in the control group. There was no corresponding effect in the intervention group (20.43 µV ± 2.38 vs. 18.45 µV ± 3, *p* = 0.483). Further, the ANOVA revealed another two-way interaction effect of “Condition” × “Electrode”, F(1,56) = 4.18; *p* = 0.046; η^2^ = 0.069. Post-hoc tests revealed a significant difference in electrode P7 between the BI and BASE condition (16.83 µV ± 1.72 vs. 18.45 µV ± 1.82, *p* = 0.013) but no effect in the P8 electrode (19.52 µV ± 2.16 vs. 20.07 µV ± 2.3, *p* = 0.867). The analysis regarding the BI-effect (BI BASE) revealed an effect of “Electrode”, with the electrode P7 (−1.61 µV ± 0.63) having a lower amplitude than electrode P8 (−0.11 µV ± 0.67), F(1,56) = 4.180; *p* = 0.046; η^2^ = 0.069. There were no other significant effects with the factors “Group” or “Appointment” (all F ≤ 0.862; all *p* ≥ 0.357).

The target P1 (t-P1) component revealed a main effect of “Condition” [F(1,56) = 14.933; *p* < 0.001; η^2^ = 0.211] where the amplitude of the BI condition was significant lower (−18.83 µV ± 2.75) compared to the BASE condition (20.96 µV ± 2.92). No other significant interactions or main effects were found (all F ≤ 3.402; all *p* ≥ 0.07). The analysis of the BI-effect (BI−BASE) revealed no significant effects with the factors “Group”, “Appointment” and “Electrode” all (F ≤ 3.402; all *p* ≥ 0.07).

The analysis of the cue N1 (c-N1) component revealed a two-way interaction effect of “Electrode” × “Group” [F(1,56) = 4.279; *p* = 0.043; η^2^ = 0.071]. Post-hoc tests showed that the amplitude of the intervention group was significant lower (−17.99 µV ± 3.93) compared to the control group (−3.1 µV ± 4.21, *p* = 0.012) at electrode P8. There was no corresponding effect at electrode P7 (−14.99 µV ± 4.06 vs. −9.52 µV ± 4.35, *p* = 0.362). No other significant interactions or main effects were found (all F ≤ 3.564; all *p* ≥ 0.064). The analysis of the BI-effect (BI−BASE) also revealed no significant effects with the factors “Group”, “Appointment” and “Electrode” (all F ≤ 3.221; all *p* ≥ 0.078).

Regarding the target N1 (t-N1) component a main effect of “Condition” was found with the amplitude of the BI condition (3.63 µV ± 2.05) being significantly lower than the BASE condition (5.23 µV ± 2.15), F(1,56) = 4.419; *p* = 0.04; η^2^ = 0.073. Another main effect of “Electrode” revealed that the amplitude of electrode P7 (2.03 µV ± 2.34) was significantly lower than electrode P8 (6.83 µV ± 2.25), F(1,56) = 5.819; *p* = 0.019; η^2^ = 0.094. No other effects were found (all F ≤ 1.384; all *p* ≥ 0.244) (Figure 4). The analysis of the BI-effect (BI−BASE) did also not reveal any significant effects with the factors “Group”, “Appointment” and “Electrode” (all F ≤ 1.384; all *p* ≥ 0.244).

The N2 component revealed no significant interactions or main effects (all F ≤ 1.622; all *p* ≥ 0.208) (Figure 5). The analysis regarding the BI-effect (BI condition−BASE condition) did also not reveal any significant effects with the factors “Group” and “Appointment” (all F ≤ 0.07; all *p* ≥ 0.792).

The analysis of the P3 component revealed a main effect of “Group” [F(1,56) = 5.508; *p* = 0.022; η^2^ = 0.09], where the amplitude of the intervention group was significant higher (26.2 µV ± 2.45) compared to the control group (17.77 µV ± 2.63). No other significant interactions or main effects were found (all F ≤ 3.116; all *p* ≥ 0.083) (Figure 6). The analysis of the BI-effect (BI−BASE) also revealed no significant effects with the factors “Group”, “Appointment” and “Electrode” (all F ≤ 2.21; all *p* ≥ 0.143).

In order to assess whether there was indeed no other effect of Group (i.e., whether the null hypothesis (H_0_) was more likely than the alternative hypothesis (H_1_), we ran Bayesian analyses [46] for all main and interaction effects of “Group”. Only the effect of “Group” in the CY-BOCS provided positive evidence for the H_1_ and the main effect of “Group” in the P3 can be interpreted as weak evidence for H_1_. All other Bayesian analyses provided positive evidence for the null hypothesis, assuming that “Group” did not modulate the data (Table 2).

We finally pursued one exploratory question of interest to see whether participants with a high symptom reduction would show an increase in the BI-effect compared to participants who do not benefit as much from using one of the apps. We performed a median split (median = 4; range: −9–23) with the differences between the pre- and post-appointment CY-BOCS scores. This allows to compare participants who benefited more vs. less from the apps to each other as well to analyze whether those, who are in the intervention group profit more specifically in terms of BI compared to participants of the control group. We therefore have another between-subject factor “Benefit” (low vs. high) regarding the ANOVA. For the RTs, the repeated measures ANOVA revealed two main effects of “Appointment” [F(1,54) = 106.257; *p* < 0.001; η^2^ = 0.663] and “Condition” [F(1,54) = 94.434; *p* < 0.001; η^2^ = 0.636] which were modulated by an interaction of “Appointment” × “Condition” × ”Benefit” [F(1,54) = 4.504; *p* = 0.038; η^2^ = 0.077].

A post-hoc repeated measures ANOVA on the BI-effect (BI condition–BASE condition) with the within subject factors “Appointment” (pre vs. post), and the two between-subject factors “Group” and “Benefit” revealed a significant interaction “Appointment” × “Benefit” [F(1,54) = 4.504; *p* = 0.038; η^2^ = 0.077] while no other effect or interaction was observed (all *p* ≥ 0.118). The interaction indicates that the amount of the BI-effect depends significantly on the presence of benefit of the app. While the BI-effect between pre-and post-appointment decreased significantly in the low-benefit group (pre-appointment: 76.9 ms ± 12.11 vs. post-appointment: 35.06 ms ± 12.82, *p* = 0.039), we observed no corresponding effect; if so, rather a trend towards the opposite direction in the high-benefit group (pre-appointment: 40.3 ms ± 10.31 vs. post-appointment: 51.82 ms ± 10.91, *p* = 0.363) (Figure 7).

## 4. Discussion

The present study was the first evaluation to investigate the efficacy of app-based mindfulness meditation training in children and adolescents with OCD using an interviewer-blind, randomized, controlled trial design. First, it was hypothesized that app-based mindfulness meditation training (intervention group) would reduce OCD related symptoms, while no symptom changes were expected while listening to an audiobook (active control group). This could not be confirmed; both groups benefitted equally in terms of OCD related symptom reduction. Overall, 77% of the intervention and 81% of control participants did benefit from using an app and had a lower CY-BOCS score after the study. The RC showed that 67% of the participants had a clinically significant reduction of OCD symptoms. The mean difference score of the CY-BOCS was 4.1 points, comparable with other studies (4.7–10.5 points) using cognitive behavioral and pharmacological treatments (see review and meta-analysis [50]). A medium effect size reflected this change. The results of the ZWIK-S and ZWIK-P also support the decreased OCD symptom severity in both groups. Additionally, dropout rates were low overall: 6 (7.9%) out of 76 participants (vs. cognitive behavioral therapy 12.7% [50]), which were all in the intervention group sample. The reasons were primarily changes from an outpatient to an inpatient hospital stay. Using app-based mindfulness meditation training or listening to an audiobook seems therefore practical as 94% in Germany of the children and adolescents from 12–19 years own a smartphone [51]. Technology-assisted treatments could represent a more efficient use of limited healthcare resources. It could also cover long waiting times for outpatient psychotherapy, as availability and related waiting times are a central problem of pediatric psychotherapeutic care (e.g., [52]). Additionally, autonomous treatment at home after first face-to-face contacts to therapists would make treatment much easier for families living in rural areas and far from treatment centers to seek help [26]. The low dropout rate in our study using an accompanied (i.e., additional, regular contacts between participants and the study team) app corroborated the feasibility of such an approach in clinical practice. Since participants reported that they used the app even after participating in the study, this can also be supported by the principle of resource activation in psychotherapy and its positive effects on the long run [53]. Yet, it may be argued that in some individuals, self-directed mindfulness did not show an effect [33]. Mindfulness interventions are used for OCD because it teaches individuals to see their thoughts as subjective, whimsical experiences. These do not necessarily reflect reality and need not be repressed or modified. In turn, individuals may develop a tolerance to obsessive thoughts, which can ease the OCD cycle [54]. However, eight weeks of practice might not have been enough for some participants, as other psychological phenomena, including the quality of mindfulness meditation training (engaging in mindfulness meditation and integration in the daily routine), influence these outcomes [55]. The purpose of mindfulness is also to teach an individual to be open-minded, non-judgmental, aware, and accepting of their current experiences [56], to facilitate the treatment of OCD [57]. However, this could have resulted in perseveration in some participants. While some may report feeling relaxed or less anxious after mindfulness practice, mindfulness exercises are not relaxation [58]. Formal mindfulness practice instructs one to observe their internal states and external experiences non-judgmentally without attempting to modify them. However, listening to an audiobook, which is considered to be a distraction, could also be considered a mindful practice since participants focused on the audiobook twice a day for eight weeks, too. Comparing a randomly assigned app-based mindfulness meditation training group with an active control group may be taken as a limitation, as it only enables us to attribute findings to using the app in combination with contact with the study team as a whole rather than to specific components within the respected app, as may have been made possible by using an additional waiting list control group. Heart rate variability and machine-led biofeedback may suggest a future integrative use to record short- and long-term effects of mindfulness states and mind wandering [50]. It must also be considered that the study treated OCD as a homogeneous condition. Given the inter-individual heterogeneity of OCD symptoms, it is suggested that it may be more useful to display OCD from a multidimensional model of symptom dimensions [59]. Therefore, various symptom dimensions in OCD may be associated with different levels of trainable mindfulness. Unfortunately, there were insufficient participants to evaluate differences based on symptom differences (e.g., washers and checkers). Nonetheless, this should be considered in future research. Moreover, the mindfulness meditation training app was not tailored to the specific OCD symptoms and might have influenced the outcome.

Besides clinical symptoms, we hypothesized that there would be significant differences between groups in the BI-effect regarding the BI paradigm [14], with the intervention group demonstrating a higher BI-effect after using an app than the (active) control group [11]. Participants of both groups showed faster RT in the BI and BASE conditions during the post-appointment compared to the pre-appointment, which suggests increased cognitive flexibility in the used task switching paradigm. However, a waitlist control group is needed in future studies to verify this effect to exclude a simple training effect. There was no significant interaction with “Group”, reflecting no difference in the change of the BI-effect between the group performing the app-based mindfulness meditation training and the group listening to an audiobook. Analogously, the absence of an interaction with “Appointment” showed that the BI-effect did not change between the pre- and post-appointment in both groups. The lack of both interaction effects has been corroborated by Bayesian analyses [46]. The neurophysiological data (EEG) did also not reveal changes, which was also corroborated by Bayesian analyses. Taken together, the use of both apps led to a symptom reduction but did not induce changes in the backward inhibition in OCD, even though we expected differences between the mindfulness meditation training and listening to an audiobook. However, compared to a previous study [11], the included groups showed a lower deficit or behavior rather comparable to a healthy control group than to an OCD group at the pre-appointment. It cannot be ruled out that the lack of changes in the BI-effect are due to a ceiling effect and that the use of the app did not modulate performance in the BI task because changes (compared to a control group) are untypically small. Moritz et al. [60] also reported no differences in BI-effects between adults with OCD and healthy controls, whereas Giller et al. [43] showed that the BI-effect of healthy adolescents was larger than in healthy adults. Thus, BI-effects change with age, which can further affect the results of an intervention in adolescence. Future studies should therefore use development-independent paradigms to investigate inhibition related effects.

An exploratory data analysis step showed that the BI-effect decreased significantly after using the app in participants who did not clinically benefit from the app (higher CY-BOCS score at the post-appointment or no change of 5 or more points). These participants already showed a higher BI-effect at the pre-appointment than participants who benefitted from using an app. If the BI level in OCD participants is already as high as in healthy controls (e.g., 92 ms, Wolff et al. [11] vs. 77 ms in the low-benefit group in this study) at baseline, there seems to be no significant difference in the BI between this OCD group and healthy controls. Thus, the proposed app usage might be more suitable for participants with an (untypically) low BI-effect at the pre-appointment (compared to healthy controls [11]).

This study is relevant for several reasons. There is scant literature on mindfulness interventions in the context of OCD symptoms, despite the need for more treatments with high acceptability and efficacy, and importantly accessibility. Mindfulness meditation training and listening to an audiobook might have the potential to be particularly useful as they can be integrated into other existing protocols, such as cognitive behavioral therapy with exposure and response prevention, or be used while waiting for a therapy place. By demonstrating the aspects of mindfulness in which individuals with OCD symptoms may be deficient, this study helped show how an accompanied app could be beneficial to this population. While large enough for the analyses in this study, larger sample sizes would allow for different methodological and statistical analyses that can answer more advanced questions, such as the chicken—egg causality between EF and OCD after mindfulness interventions and between OCD symptomatic and comorbidities, as mentioned before. In addition, this study used only the severity of OCD as an indicator. However, as OCD patients exhibit various comorbidities, further studies should analyze various indicators [61]. Future studies should also consider follow-up appointments to identify long-term effects.

## 5. Conclusions

In conclusion, the present study findings indicate that app-based mindfulness meditation training or listening to an audiobook decreases symptom severity in pediatric OCD with a RC of 67%. However, mindfulness meditation training does not have a specific effect on symptom severity compared to listening to an audiobook, as expected. There are several areas identified by this study for future investigations. First, this study furthers our understanding of EF processes in pediatric OCD after a mindfulness meditation training and is worthy of future research as EF differences compared to healthy controls are an evident trait of OCD [62,63,64]. The baseline BI-effect should be considered as an intra-individual component and could predict the study outcome in regard to symptom severity accordingly. Furthermore, participants with a high vs. those with low symptom severity reduction could show different BI-effects regardless of mindfulness meditation training or listening to an audiobook.

Second, compared to clinicians, researchers have not yet begun to investigate mindfulness meditation training for the treatment of pediatric OCD or to examine potential mechanisms [58]. Therefore, the present findings provide a basis from which mindfulness interventions can be further explored in pediatric OCD. This may help to clarify whether particular aspects of OCD, such as certain symptom dimensions or psychological features (e.g., impaired inhibition), have more direct relationships to the mindfulness construct than a more global measure of OCD symptomatology. Experimental work could also help explain the nature of different mindfulness interventions in individuals with OCD, such as tailored programs for specific OCD symptoms and comorbidities. Future studies should include a waiting list control group do differentiate between the reported findings.

## Figures and Tables

**Figure 1 jcm-12-02486-f001:**
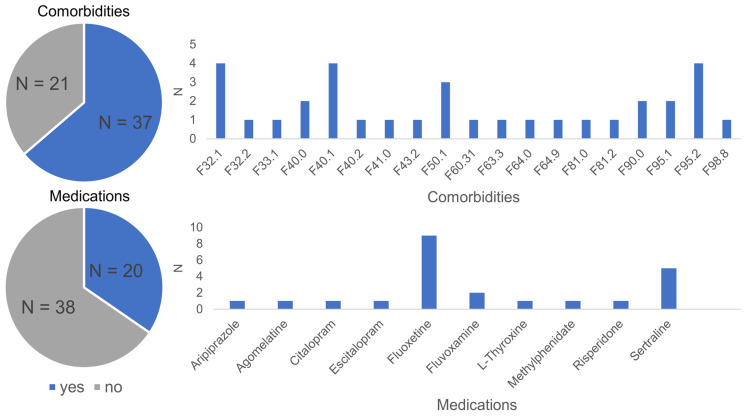
Comorbidity and medication status of N = 58 included participants.

**Figure 2 jcm-12-02486-f002:**
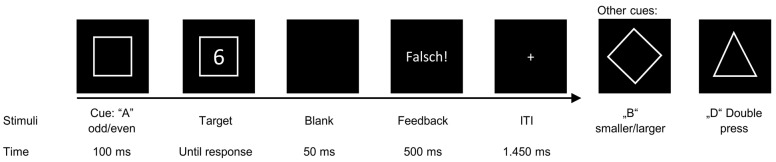
The outline of the backward inhibition paradigm including the experimental logic is presented. Each trial begins with the presentation of a cue in the center of the screen. Cue “A” is a square and indicates to decide whether the target is odd or even. Cue “B” is a diamond and indicates to decide whether the target is smaller or larger than 5. Cue “D” is a triangle and indicates a double-press regardless of the target. After 100 ms, the target stimulus is presented within the cue stimulus until a response is made. During the inter-trial-interval (ITI) of 2.000 ms, there is feedback of 500 ms for incorrect trials with the German word “Falsch!” (English: “Incorrect!”). There is no feedback in correct trials.

**Figure 3 jcm-12-02486-f003:**
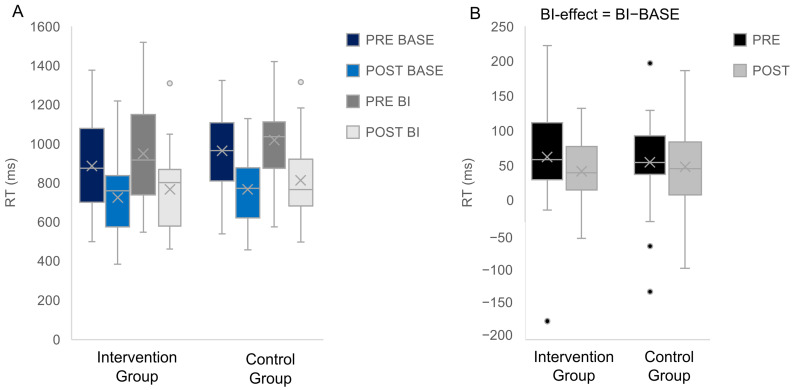
Reaction time data for the BI and the BASE condition in the intervention and the control group are presented (**A**). The BI-effect (i.e., BI condition−BASE condition) for the two groups is shown (**B**). Within each box, horizontal grey lines denote median values; boxes extend from the 25th to the 75th percentile of each condition’s distribution of values; whiskers above and below the box indicate the 10th and 90th percentiles (i.e., the most extreme values within 1.5 interquartile range of the 25th and 75th percentile of each condition); dots denote observations outside the range of adjacent values.

**Figure 4 jcm-12-02486-f004:**
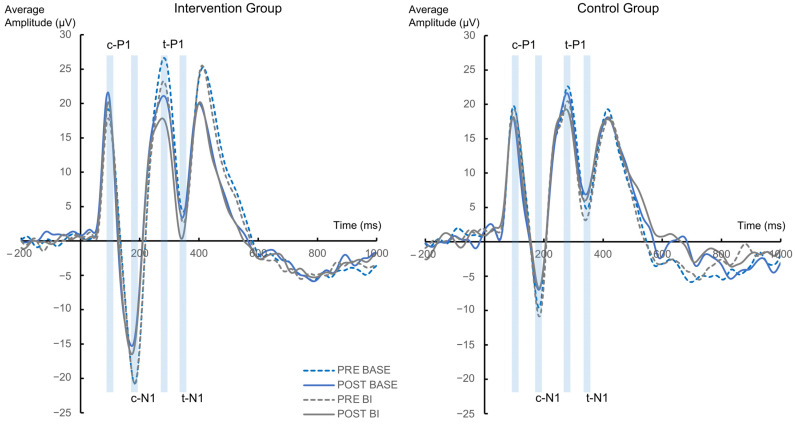
Cue-P1 (c-P1), cue-N1 (c-N1), target-P1 (t-P1) and target-N1 (t-N1) ERPs evoked by the cue or target stimuli pooled across electrode P7 and P8 are presented for the intervention group and control group separately. Time point zero denotes the onset of the cue; the target stimulus was added to the visual array 100 ms later.

**Figure 5 jcm-12-02486-f005:**
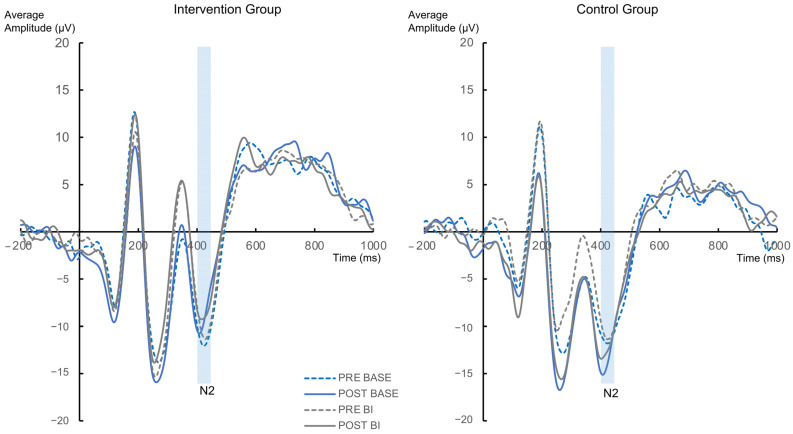
The N2 ERPs at electrode Cz are presented for the intervention group and control group separately. Time point zero denotes the onset of the cue.

**Figure 6 jcm-12-02486-f006:**
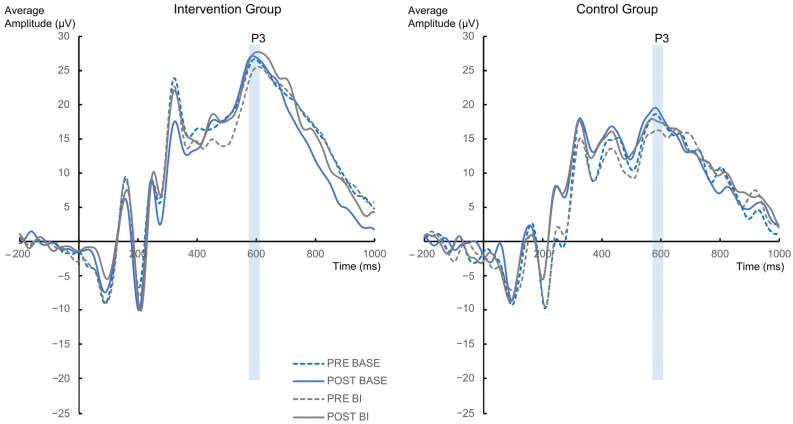
The P3 ERPs pooled across electrode PO1 and PO2 are presented for the intervention group and control group separately. Time point zero denotes the onset of the cue.

**Figure 7 jcm-12-02486-f007:**
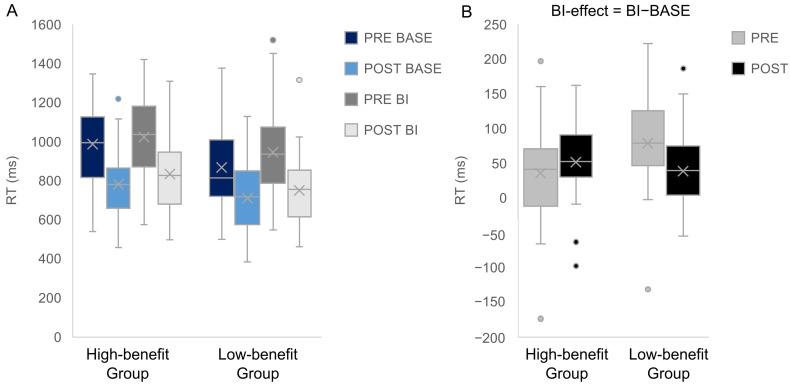
Reaction time data for the BI and the BASE condition in the high-benefit group and low-benefit group are presented (**A**). The BI-effect (i.e., BI condition/BASE condition) for the two groups is shown (**B**). The groups were defined by a median split (median = 4) regarding the differences between the pre- and post-appointment CY-BOCS scores. Within each box, horizontal grey lines denote median values; boxes extend from the 25th to the 75th percentile of each group’s distribution of values; whiskers above and below the box indicate the 10th and 90th percentiles (i.e., the most extreme values within 1.5 interquartile range of the 25th and 75th percentile of each group); dots denote observations outside the range of adjacent values.

**Table 1 jcm-12-02486-t001:** Demographics and clinical characteristics of the final sample (N = 58).

	Intervention Group (N = 31)		Control Group (N = 27)		Comparison between Groups			
	M (SD)	Range	M (SD)	Range	t	df	*p*	95% CI
Age in years	15.75 (2.17)	11.66–19.59	15.29 (1.99)	10.73–18.97	0.85	56	0.399	[−0.64–1.57]
IQ ^a^	106.34 (14.09)	85–139	99.39 (13.7)	79–136	1.898	56	0.063	[−0.39–14.29]
**CY-BOCS ^b^**								
Obsessions PRE	9.19 (4.22)	0–19	7.93 (3.58)	1–17	1.223	56	0.017	[−0.81–3.35]
Obsessions POST	7.32 (3.54)	0–13	6.00 (3.14)	0–12	1.495	56	0.141	[−0.45–3.1]
	t(30) = 2.862; *p* = 0.008		t(26) = 3.366; *p* = 0.001					
Compulsions PRE	9.97 (3.29)	5–18	7.78 (3.49)	3–16	2.458	56	0.018	[0.41–3.98]
Compulsions POST	7.84 (3.23)	0–13	5.48 (3.67)	0–12	2.603	56	0.012	[0.543–4.17]
	t(30) = 3.298; *p* = 0.003		t(26) = 4.395; *p* < 0.001					
Total score PRE	19.48 (7.28)	8–37	15.33 (5.84)	7–24	2.369	56	0.021	[0.64–7.66]
Total score POST	15.16 (6.36)	0–25	11.48 (6.51)	0–23	2.175	56	0.034	[0.291–7.07]
	t(30) = 3.776; *p* < 0.001		t(26) = 4.386; *p* < 0.011					
**ZWIK-S ^c^**								
PRE	43.65 (29.82)	2–123	39.52 (26.95)	4–101	0.55	56	0.585	[−19.92–19.17]
POST	37.28 (27.16)	5–113	26.48 (18.53)	5–77	1.725	54	0.09	[−1.75–23.34]
	t(28) = 2.255; *p* = 0.016		t(26) = 3.237; *p* = 0.003					
**ZWIK-P ^d^**								
PRE	30.1 (20.04)	4–84	34.52 (16.52)	8–72	−1.018	56	0.313	[−14.65–4.77]
POST	23.77 (15.75)	3–59	26.37 (18.81)	7–88	−0.569	55	0.572	[−11.78–6.57]
	t(29) = 2.129; *p* = 0.04		t(26) = 2.09; *p* = 0.047					
Total minutes App used	813.16 (278.62)	268–1231	1081.48 (283.57)	166–1436	−3.628	56	<0.001	[−416.46–−120.18]
	**N (%)**		**N (%)**		**X^2^**	**df**	** *p* **	
Male	13 ^e^ (42)		12 (44)		0.037	1	0.847	
Comorbidity	12 (39)		9 (33)		0.181	1	0.671	
Medication	12 (39)		8 (30)		0.527	1	0.468	

^a^ Processing speed of the Zahlen-Verbindungs Test (ZVT) (English: number-connection test) was used to estimate the intelligence quotient (IQ) [39] ^b^ Children’s Yale-Brown Obsessive-Compulsive Scale with a total severity score range from 0 to 40 (CY-BOCS [36]; German: Steinhausen, 2007). CY-BOCS: Total score between 8 and 15 is considered mild, 16–23 moderate, 25–30 severe, and over 30 extreme (Bossert-Zaudig & Niedermeier [40]). ^c^ Zwangsinventar (English: obsessive-compulsive inventory) self-report [41] (ZWIK-S): with a total score range from 0 to 144, N = 56 participants included ^d^ Zwangsinventar (English: obsessive-compulsive inventory) parent-report [41] (ZWIK-P): with a total score range from 0 to 144, N = 57 participants included ^e^ gender: 2 female-to-male transgender participants included.

**Table 2 jcm-12-02486-t002:** Bayesian analyses for each effect of the factor “Group”.

	BF	P_BIC_(H_0_|D)	P_BIC_(H_1_|D)
**Symptom data**			
**CY-BOCS**			
Main effect Group	0.34	0.25	0.75
Interaction Group × Appointment	7.23	0.88	0.12
**ZWIK-S**			
Main effect Group	3.66	0.79	0.21
Interaction Group × Appointment	3.56	0.78	0.22
**ZWIK-P**			
Main effect Group	5.16	0.84	0.16
Interaction Group × Appointment	7.02	0.88	0.12
**Behavioral data**			
**Hit RT**			
Main effect Group	4.07	0.80	0.20
Interaction Group × Appointment	5.38	0.84	0.16
Interaction Group × Appointment × Condition	8.57	0.90	0.10
**Hit Accuracy**			
Main effect Group	7.5	0.88	0.12
Interaction Group × Appointment	6.37	0.86	0.14
Interaction Group × Appointment × Condition	2.05	0.67	0.33
**Neurophysiological data**			
**c-P1**			
Main effect Group	7.11	0.88	0.12
Interaction Group × Appointment	4.83	0.83	0.17
Interaction Group × Appointment × Condition	12.02	0.92	0.08
**t-P1**			
Main effect Group	7.47	0.88	0.12
Interaction Group × Appointment	5.77	0.85	0.15
Interaction Group × Appointment × Condition	12.91	0.93	0.07
**c-N1**			
Main effect Group	1.27	0.56	0.44
Interaction Group × Appointment	7.24	0.88	0.12
Interaction Group × Appointment × Condition	1.65	0.62	0.38
**t-N1**			
Main effect Group	6.32	0.86	0.14
Interaction Group × Appointment	4.96	0.83	0.17
Interaction Group × Appointment × Condition	3.96	0.8	0.2
**N2**			
Main effect Group	5.41	0.84	0.16
Interaction Group × Appointment	3.33	0.77	0.23
Interaction Group × Appointment × Condition	11.82	0.92	0.08
**P3**			
Main effect Group	0.5	0.33	0.67
Interaction Group × Appointment	7.55	0.88	0.12
Interaction Group × Appointment × Condition	9.36	0.9	0.1

BF = Bayes factor, P_BIC_(H_0_|D) = posterior probability for the null hypothesis (i.e., the probability of the null hypothesis, given the obtained data); P_BIC_(H_1_|D) = posterior probability for the alternative hypothesis. P(H|D) values of 50–75% can be interpreted as weak evidence, values of 75–95% can be interpreted as positive evidence, values of 95–99% can be interpreted as strong evidence, and values above 99% can be interpreted as very strong evidence for a given hypothesis [49].

## Data Availability

Data presented in the study can be found under the following link: https://osf.io/v5tqp/?view_only=698a87c3522844d8af5915e39543b563 (accessed on 19 December 2022).

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
