# Peer review of "App-Based Mindfulness Meditation Training and an Audiobook Intervention Reduce Symptom Severity but Do Not Modify Backward Inhibition in Adolescent Obsessive-Compulsive Disorder: Evidence from an EEG Study"

_jcm, 2023, doi:10.3390/jcm12072486_

Round 1

Reviewer 1 Report

Dear Editor, dear Authors,

Thank you for letting me reviewed this paper. This paper aim to measure the efficacy of a smartphone app delivering a mindfullness based therapy on OCD symptom severity of adolescents OCD patients, and then to demonstrate the effect of a such app on a supposed measure of the cognitive flexibility (measured through the behavior and through eeg).

As a whole, this paper is mainly characterized by its lack of results and by the fact that the authors tried to hide this.

First of all, the authors are way too much affirmative claiming that their intervention is effective : indeed, following their results, it is clear that their app is not more effective than the control condition used (listening to an audiobook). The authors recognized this but then claimed that finally, listening a story (the control condition supposed to have no effect) or doing mindfulness is effective on OCD symptom severity. It is difficult to believe that listening a book is effective on OCD severity, and my clinical experience does not go that way… Basically, the better way to show effectiveness of an intervention is to compare the mean (or the ranks) between such an intervention and a control condition. Here, the authors did not do that, they used an ANOVA (why not) but they did not mention the result of this ANOVA that concerned the efficacy of their method ! Instead, they use an index I never saw before…. but only for the  measure of the OCD severity through the CYBOCS. Then, weirdly, such an index disappear for the other measures of the OCD severity.

Therefore, as they mentioned themselves the audiobook condition as a control condition, they can not finally say that mindfullness is as effective as the control condition, therefore the control condition is also effective (especially in the absence of a « waiting-list » group).

Then, for the behavioral and electrophysiological measure of their intervention, the authors also recognized that both the control and the mindfulness condition led to the same results, results probably due to an effect of training.

Finally, the author, rather that raising the limitations of their study before to conclude (limitations are numerous, it is even not sure that OCD patients are really OCD patients : « Outpatients with a confirmed OCD diagnosis (ICD-10 diagnoses F42.0, F42.1, F42.2, F42.8 or F42.9) and individuals with a suspected OCD and a CY-BOCS score ≥ 8 points were included (see Table 1). »), they prefered to highlight the strengths of their paper…

I had other concerns but to my opinion, the very first thing should have been to present this work as having no positive results rather than trying to hide this and to highlight strengths rather than limitations.

Reviewer 2 Report

Although the manuscript for this paper is complete, the proposed content is not clearly presented. In order for it to be considered for high-quality publication, many improvements must be made, particularly some figures and tables’ format which do not clearly convey the high quality of this domain to readers. In my opinion, significant improvement in the writing of the content is necessary for publication.

Specific comments are as follows:

(1)    In Materials and Methods section, sub-section title should be added with 2.1 Participants, 2.2 Intervention and procedure, 2.3 Backward Inhibition Task and so on.

(2)    In Figure 3B, the reaction time data for the BI-effect present negative results. I think that the negative value represents the difference between the time the stimulus was presented and the time of the response, which may be slightly before the stimulus was actually presented. However, these negative values are usually considered to be within the margin of error and are not meaningful in the analysis of reaction time data. Or could you explain the results?
